# MULTI-CONCEPT EDITING USING TASK ARITHMETIC

## ABSTRACT

Model owners often wish to introduce new capabilities into their trained models or remove undesired ones. Task Vectors (TVs) present a promising new approach to editing models after training, allowing simple and controllable addition of new capabilities to the model and the removal of undesired ones. But what happens when the model owner wants to change multiple capabilities?

In this work, we study the interactions of task vectors in a multi-edit setting for image classifiers and diffusion models. We start by quantifying the overall model degradation induced by applying many specific TVs simultaneously. We show that the overall model performance degrades rapidly as the quantity of TV edits increases. Finally, we explore different ways to mitigate this degradation and present an adaptive method to select the most relevant TVs to apply to a diffusion model during inference. Our technique achieves a $94.6\%$ ROC AUC in identifying the correct TV, enabling the effective integration of multiple TV edits while significantly mitigating quality degradation.

## 1 INTRODUCTION

As advances in machine learning increasingly rely on large foundation models trained by entities with substantial resources, the need to adapt these models to various end-user preferences is growing (Zhuang et al., 2020). One straightforward way to adapt a foundation model is to fine-tune it on a relevant dataset directly. However, this approach may suffer from issues including privacy concerns (Yu et al., 2021), stability (Wortsman et al., 2022b; Mitchell et al., 2021), and computational resources (De Cao et al., 2021).

One popular alternative to fine-tuning are *task vector edits* (TV edits), which perform algebraic operations or task arithmetic, such as addition or subtraction directly on the model weights. The *task vector* (TV) is a learned set of weights representing the difference between the pre-trained model weights $\theta_0$ and a fine-tuned model weights $\theta_{\text{ft}}$ (Ilharco et al., 2022a). For example, to reduce the likelihood of a model generating pictures in the style of Vincent Van Gogh, a model owner might perform a TV edit, subtracting a task vector learned from Van Gogh's images $\tau_1$ from the base model weights to yield a sanitized model $\theta_{TV}$:

$$\theta_{TV} = \theta_0 - \alpha_1 \tau_1 \tag{1}$$

TV edits attain stability and robustness not achievable by other methods (Tsai et al., 2023; Pham et al., 2023; 2024). In addition, TVs finetuned on a narrow task may allow for better control on the generation of specific attributes (Gandikota et al., 2023). Even so, in the pursuit of better *target task performance* on the new behavior, task vector edits can sometimes impair the quality of unrelated generations, also known as *control task performance*. The trade-off between the target task performance and control task performance has received considerable attention in prior work on single TV applications (Gandikota et al., 2023; Pham et al., 2024). Additionally, a growing line of research focuses on combining TVs representing broad fine-tunes of the model (also known as model merging), which should generally improve the overall performance and not degrade it (Yadav et al., 2024; Xu et al., 2024; Matena & Raffel, 2022). Yet, TVs aimed at a specific class or concept generally degrade the control task performance.

In this work, we raise a new, related question; what happens when the model owner wants to change multiple capabilities, requiring multiple TV edits together? For example, erasing the ability to generate

or recognize many different human identities (Zehavi & Shamir, 2023), removing copyrighted styles (Pham et al., 2023), or precisely controlling generated attributes (Gandikota et al., 2023). We study the interactions of task vectors in this multi-edit setting, and uncover an unusual phenomenon; multiple TV edits interact not only with the model itself but also *with one another* in their effect on the control task performance. We term this behavior *multi-task interactions*.

We find that pairwise-task interactions of TVs can be modeled as a combination of two contrasting regimes: highly colinear TVs tend to have *linear* multi-task interactions, meaning the control task accuracy decrease is a function of the total magnitude of the applied TVs[1]. Less similar TVs (which tend to be induced by unrelated tasks Ilharco et al. (2022a)) usually have non-linear multi-task interactions: editing with two different vectors will be less harmful to the control task than using a single TV with the total magnitude (see Fig.1). Going beyond pairwise interactions to very large number of simultaneously applied TVs, we find that the linear interactions dominate; accuracy degrades *linearly* with the total magnitude of the vectors being subtracted. We offer a simple theoretical explanation for this observation.

Next, we identify and evaluate various natural approaches to mitigate this accumulated degradation in model accuracy. We test the following methods: (i) merging Task Vectors with a non-linear merging algorithm developed for model-merging (non concept-specific TVs), (ii) learning a per-TV magnitude for a better erasure/control trade-off, (iii) training a single joint Task Vector for multiple concepts, and (iv) using the Neural Tangent Vector TV method (Ortiz-Jimenez et al., 2024). We find that all the above methods fall short in reducing the model degradation to acceptable levels when applying a large number of edits.

Finally, we propose a technique to choose at inference time which TVs to use for a diffusion process. As different TVs are tuned to edit different concepts, most generations do not require a large number of concurrent subtractions. Motivated by this intuition, we investigate whether we can determine which TVs to apply to a diffusion model with a given prompt only by analyzing the effect of the TV edit on the generated image. We determine that applying a TV in the middle of the denoising process allows us to quantify its relevance to a given prompt. We flesh this idea out into a technique that applies TVs only when they are relevant to avoid unnecessary model deterioration.

**Our Contributions.**

- We conduct an initial study of how multi-task-vector interactions affect the control task performance of a model.
- We test existing methods for mitigating the control task performance decline under multi-task-vector edits, and find that all methods fall short at sufficient scale.
- We propose a novel inference-time solution to adaptively applying only the relevant Task Vectors to a given prompt.

## 2 BACKGROUND

**Merging Model Edits.** There is a growing research interest in methods that take a few models, trained or fine-tuned separately, and combine them post-training into a single model enjoying the strengths of all the individual fine-tunes. While some approaches individually run each model and combine the model outputs in one of several ways Dietterich (2000); Ovadia et al. (2019); Gontijo-Lopes et al. (2021), other methods combine the model parameters themselves Chung (1954); Wortsman et al. (2022a); Yadav et al. (2024). The Task Vector method combines model parameters by fine-tuning a few different models from the same checkpoint and averaging the differences in parameters accumulated in each model along the fine-tuning process. While averaging models at parameter space may sound unintuitive, it was shown to be semantically meaningfulIlharco et al. (2022a), and a few techniques were suggested to better optimize it Ortiz-Jimenez et al. (2024); Yadav et al. (2024); Goddard et al. (2024). Yet, these techniques were mostly focused on the setting where each TV was trained to add to the model a relatively broad capability (e.g., better general generation quality or robustness). Less focused was directed to TV for specific narrow concepts, and concept erasure.

---

[1]Note that this kind of linearity is distinct from previous work, which has focused on linearizing the task vectors themselves Ilharco et al. (2022b;a); Ortiz-Jimenez et al. (2024)

**Diffusion Models.**  Diffusion models Sohl-Dickstein et al. (2015); Ho et al. (2020) are a class of generative models that learn to sample from a distribution using a Markovian denoising process. In the forward diffusion process, Gaussian noise is gradually added to an input image $\mathbf{x}_0$ for $T$ timesteps to yield a final noise latent $\mathbf{x}_T$. The model learns the reverse diffusion process, which, given a latent $\mathbf{x}_t$ at a timestep $t$, predicts the residual noise $\boldsymbol{\epsilon}_t = \mathbf{x}_t - \mathbf{x}_{t-1}$. At inference time, a random Gaussian noise tensor $\mathbf{x}_T$ is sampled and passed through the reverse process for $T$ timesteps to yield the final data $\mathbf{x}_0$. Latent Diffusion Models (LDMs) Rombach et al. (2022) reduce the memory footprint of diffusion models by performing the denoising process in a latent space learned using an autoencoder.

**Task Vector Edits to Diffusion Models.**  Among many methods suggested to address model editing for diffusion models, we focus on Task Vectors, as they are most suited to study the interaction between tasks.  Practically, Task Vectors have been used in diffusion models to achieve better controllability Gandikota et al. (2023) and concept erasure Pham et al. (2024); Liu et al. (2024).We acknowledge the vast literature covering other editing method for diffusion models Orgad et al. (2023); Bau et al. (2020); Croitoru et al. (2023); as well as on applying task vectors to edit other types of models Hendel et al. (2023); Ramesh et al. (2024); Hojel et al. (2024). However, as our primary focus in the paper is on conceptual questions, we adhere to Task Vector edits applied to Diffusion Models and classifiers.

## 3 Task Vector Interactions

Our study begins with a simple question:

> How do multiple TV edits performed together affect the model performance?

Our practical motivation for studying multi-task interaction is applying multiple Task Vectors to a single model simultaneously. Yet, there is also a deeper scientific motivation for exploring this question. Ideally, Task Vectors aim to represent a single edit direction (e.g., happy vs. sad) of the model while mostly keeping other edit directions unaffected (e.g., outdoor vs. indoor) (Ilharco et al. (2022a)). Examining the interaction of multiple task vector edits allows us to better inspect the extent to which TV are non-interfering, and study the interference caused by the combination of different tasks.

### 3.1 Pairwise Task Interactions

The simplest kind of multi-task interaction is a *pairwise* interaction; two task vectors $\tau_1, \tau_2$ are applied to one model $\theta_0$ with strengths (amplitudes) $\alpha_1, \alpha_2$, generating a model $\theta_{TV}$:

$$\theta_{TV} = \theta_0 + \alpha_1 \cdot \tau_1 + \alpha_2 \cdot \tau_2 \tag{2}$$

We follow previous works by evaluating a pre-trained CLIP-based classifier model as $\theta_0$, our base model; and extend it to examining the unet of a Stable diffusion model in App.C. We explore a large variety of classification tasks for the CLIP-based model, and different artistic styles and objects for the stable diffusion model. Finally, we plot the control task performance (classification or generation of unrelated concepts) as a function of the edit strengths of the two vectors, $\tau_1, \tau_2$ (the magnitudes are noted as $\alpha_1, \alpha_2$ serve as the axes of the control task performance heatmap). While varying the control task reveals diverse interaction patterns, the pair-wise interaction effects on standard tasks mostly fall into two categories, see Fig. 1 (see Appendix for implementation details and more results). We note these categories as *Linear interactions* and *non-linear interactions*.

**Linear interactions.**  In this type of TV interaction the degradation effect of using one amplitude, $\alpha_1$, for one TV, and a second amplitude, $\alpha_2$, for another, is similar to the effect of using the sum of amplitudes $(\alpha_1 + \alpha_2)$ with one of them (see upper panel of Fig. 1). One simple such case is the interaction between a TV and itself ($\tau_1 = \tau_2$). In this case, Eq. 2 trivially becomes:

$$\theta_{TV} = \theta_0 + (\alpha_1 + \alpha_2) \cdot \tau_1 \tag{3}$$

Therefore, the performance degradtion trivially becomes a function of $(\alpha_1 + \alpha_2)$. This kind of interaction is also expected in highly correlated tasks that move the similar weights in the same direction.

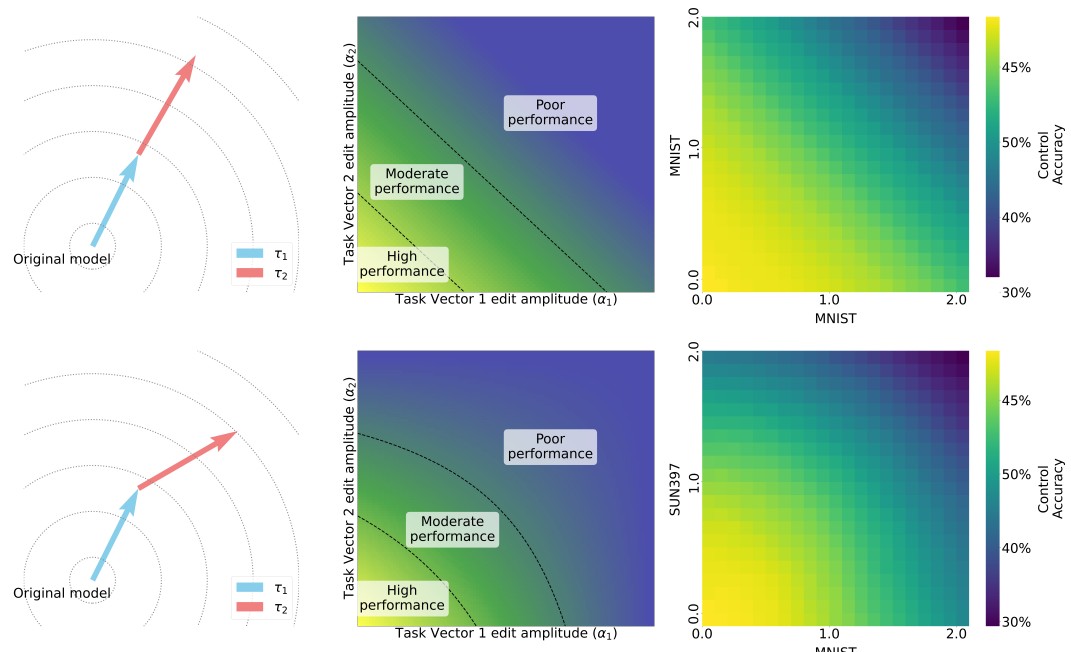

Figure 1: *Illustration of model performance influenced by task vector editing for two scenarios* **(Top panels)** *Linear interaction* **(Bottom panels)** *Non-linear interaction.* **(Left panels)** *visualize the total magnitude of two task vectors, $\tau_1$ and $\tau_2$, with different angles between them.* **(Middle panels)** *schematically illustrate equi-performance lines in the space of amplitudes of the applies TVs $(\alpha_1, \alpha_2)$ of possible TV interactions, highlighting the performance in different regions.* **(Right panels)** *feature heatmaps displaying the empirical control task performance; namely, ImageNet classification accuracy, when the CLIP backbone was edited with TV associated with different tasks noted as the axes titles. High performance corresponds to yellow areas, moderate performance to green, and poor performance to dark blue. More similar plots can be found in App.8*

**Non-linear interactions.** non-linear interactions are interactions where the combined effect of a few TVs on the control accuracy is smaller than that expected according to the individual effects of each TV. In the non-linear interactions regime, the amount of model degradation is a non-linear function $f(\alpha_1, \alpha_2)$ of the edit strengths. Intuitively, conceptually unrelated TVs will have a small shared components, and will be mostly orthogonal to one another . Therefore, we can expect the joint vector magnitude to be effectively smaller than the magnitude addition; as the sum of non-co-linear vectors (Fig.1, leftmost figure).

Using this intuition, we suggest a simple toy model to explain a variety of interactions. For this model, we consider each TV as composed of two components: (i) a joint component $\boldsymbol{\mu}$, related to the semantically shared properties of the tasks used to train the two TVs (e.g., the MNIST Deng (2012) and SVHN Netzer et al. (2011) classification are likely to share such a component as both tasks require reasoning about digits, see App.Fig.8). (ii) An uncorrelated component, related to parameter changes idiosyncratic to fine-tuning procedure (e.g., different low level color features which are unrelated even between Mnist and SVHN). Therefore, we model these components as coming from a random distribution with a covariance matrix $\mathcal{N}(0, \boldsymbol{\Sigma})$. Taken together, we model our TVs as follows:

$$\tau_1, \tau_2 \sim \mathcal{N}(\boldsymbol{\mu}, \boldsymbol{\Sigma}) \qquad (4)$$

In that setting, a combined TV can be represented as its own Gaussian drawn from the following distribution:

$$\alpha_1 \tau_1 + \alpha_2 \tau_2 \sim \mathcal{N}((\alpha_1 + \alpha_2)\boldsymbol{\mu}, (\alpha_1^2 + \alpha_2^2)\boldsymbol{\Sigma}) \qquad (5)$$

We consider two similar TVs as having a high common mean $\mu$ and a small variance $\Sigma$ compared to this mean ($\mu >> \sqrt{|\Sigma|_\infty}$). Therefore, the term $(\alpha_1 + \alpha_2)\boldsymbol{\mu}$ dominates over the covariance term, making this a mostly additive interaction. However, a small semantic similarity corresponds to the

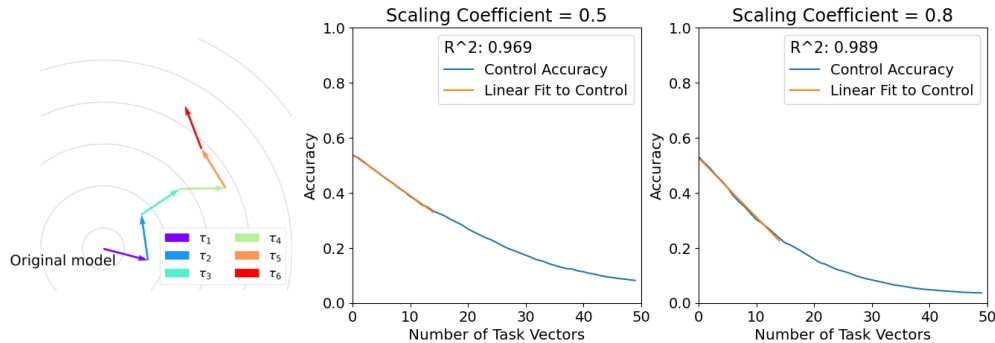

Figure 2: ***The control task deteriorates linearly with increasing the amount of subtracted TVs.*** *(**Left**) Illustration of many TV addition when they share a common average component, projected to two dimenstions. (**Middle, Right**) Control task accuracy (50-classes CIFAR classification) of a CLIP backbone as function of the number of TV edits applied with a fixed magnitude each. When editing an increasing number of TVs, the control task performance linearly degrades up to a significant amount of vectors. We examine two different scaline coefficients ($\alpha_C$) depicted in the title of each figure.*

covariance dominating the joint component of the vectors ($\sqrt{|\Sigma|_\infty} >> \mu$). In this case, the standard deviation term dominates, and its scale $\sqrt{\alpha_1^2 + \alpha_2^2}$, grows sub-linearly. We show in Fig. 8 (panel D) an empirical interaction of random TVs (averaged over a few seeds to reduce random noise).

One way to quantify this relationship is by examining the actual angle $\phi$ between the given vectors (App.Fig. 8). However, vectors in very high dimensions tend to be nearly orthogonal, and the connection between image semantics and model weights is implicit. Therefore, a finer way to study the similarity between given TV pairs is to look at the number of model layers with an internal angle above a fixed threshold $\phi_i > \phi_t$. Setting $\phi_t = 75°$, we find that the intuitively correlated tasks have fewer layers whose angle is above $\phi$, while the less related task pairs have many such layers. Comparisons of the angles using this model can be found in the App.Fig.8.

## 3.2 SCALING EFFECTS UNDER MULTIPLE TASK INTERACTIONS

Having gained insight into pairwise Task Vector interactions, we turn to study model degradation when editing with a large number of TVs at once:

$$\theta_{TV} = \theta_0 + \sum_{i=1}^{N} \alpha_i \cdot \tau_i \tag{6}$$

As plotting the model performance as a function of $\{\alpha_i\}$ no longer fits on a 2D heatmap, we turn to another evaluation method. We use a constant magnitude $\alpha_c$ and add many Task Vectors with the same magnitude. We treat here erasing single CIFAR-100 classes as our target task, and classification accuracy on the last 50 CIFAR-100 classes as our control task. Using this setup leaves us with 50 task vectors to study, one for each of the first 50 classes. The results can are shown in Fig. 2.

We can see that up to a significant number of vectors (15 TVs), the accuracy degradation is linear as a function of the number of subtracted TVs. The degradation cannot, of course, remain linear for an arbitrarily high number of subtractions as the accuracy is bounded from below by zero. Yet, the close linear fit provides a strong indication that the *linear interaction* pattern is dominant over the *non-linear interaction* pattern. Our simple mathematical model suggests a simple explanation for the phenomenon.

Assuming even a small shared average component $\mu$ between any given pair of vectors, we may describe the TVs as drawn from a distribution denoted as follows:

$$\tau_i \sim \mathcal{N}(\boldsymbol{\mu}, \boldsymbol{\Sigma}) \tag{7}$$

Summing many TVs, all with magnitude $\alpha_c$ we have:

$$\sum_{i=1}^{N} \alpha_c \cdot \tau_i \sim \mathcal{N}(\alpha_c \cdot \boldsymbol{\mu} \cdot N, \alpha_c^2 \cdot \boldsymbol{\Sigma} \cdot N) \tag{8}$$

In the case of many such TVs, the mean term $\boldsymbol{\mu}$ grows linearly while the standard deviation grows as square root $\sqrt{N}$. Therefore, for large values of $N$ the mean dominates over the standard deviation, and the linear interaction pattern is dominant.

## 4 MITIGATING CONTROL TASK PERFORMANCE DEGRADATION VIA MULTI-TASK ARITHMETIC

In the last section, we saw that the control task performance impacts accumulate linearly at scale, making it difficult to apply multiple TV edits. In this section, we investigate whether this degradation can be mitigated. We survey four solutions using either existing methods or simple modifications to the TV techniques, and conclude that none of them work well enough to allow the practical application of TV-based concept erasure at scale.

**Non-Linear TV Combination.** The standard way to combine TVs is using simple algebric vector addition in the weight space. It might be the case, though, that other notions of combination such as non-linear TV combinations may better preserve the control task accuracy. In fact, such methods have already been proposed for model merging, where edits aim to represent global improvement rather than changes to a specific concept or class. In the model merging case, we aim to reap the benefits of all the fine-tuned instances together Wortsman et al. (2022a). Yet, we can also evaluate these techniques for narrow tasks vectors such as ones finetuned on a single class. We therefore evaluate 4 alternatives for parameter-wise TV combination: *(i) Linear* - regular linear addition of the model weights (Baseline). *(ii) Sparse* - we sparsify the TVs such that each vector contains only the weights of the top $p$ percentiles of TV parameters, sorted by magnitude, then add the sparse TVs linearly as in the standard method. The precentile $p$ is varied to inspect different points on the control-target tradeoff. *(iii) Median* - Similar to *Linear* but taking the median rather than the sum of each of the parameters. A global magnitude factor can be used to better explore the trade-off between the control task performance and the target task performance. *(iv) Tie merging Yadav et al. (2024)* - A leading method for combining positive TV edits. We find that all TV combination methods give a similar control-target trade-off (Fig. 3, App.D for implementation details).

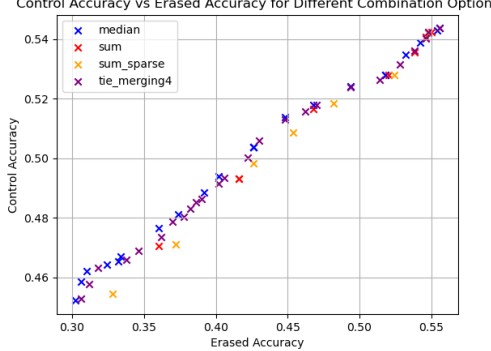
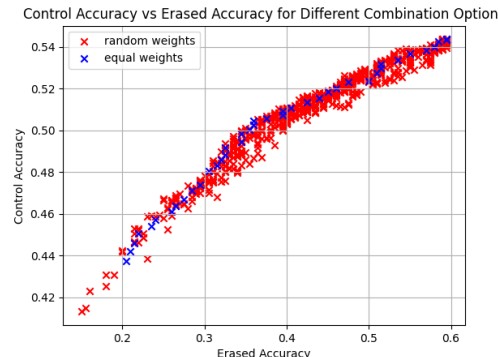

Figure 3: ***Four TV combination methods give a similar control-target tradeoff.*** *We wish to get better control accuracy for a given target task (Here, concept erasure. Lower is better.) performance trade-off. Yet, different TV combinations methods performance are giving a similar control-target trade-off curve.*

Figure 4: ***Per-TV edit strengths give a similar control-target tradeoff.*** *We plot the control task accuracy and target task (Here, concept erasure. Lower is better.) performance tradeoff, once with equal magnitude for each TV (Blue) and once when randomizing a different magnitude for each TV (Red).*

**Learnable Task Vector weights.**    Applying all Task Vectors with the same magnitude is usually enough to ensure the edit is applied for all concepts, assuming the magnitude is large enough. Yet, since the different TV edits may affect one another, the trade-off between the control task performance $\text{Acc}_{\text{ctrl}}$ and the target task performance $\text{Acc}_{\text{target}}$ could potentially benefit from better optimization of the TV weights. We wish to find optimal weights, and formulate the problem as follows:

$$\arg\max_{\alpha} \text{Acc}_{\text{ctrl}}(\theta_{TV}; \alpha) + \lambda \|\text{Acc}_{\text{target}}(\theta_{TV}; \alpha)\|_2^2 \qquad (9)$$

Here $\text{Acc}_{\text{target}}(\theta_{TV}; \alpha)$ is the concept erasure performance (minus accuracy) on all of the target tasks, and $\text{Acc}_{\text{ctrl}}(\theta_{TV}; \alpha)$ is the accuracy of the backbone classifier on 50 unrelated classes. The parameter $\lambda \in \mathbb{R}$ controls the relative importance of the control task performance, allowing us to inspect the control-target trade-off. Using this loss function we aim to find a per task vector $(\alpha_i)$ vector that can minimize the erased accuracies while preserving control accuracy as much as possible.

As optimizing this function with stochastic gradient descent did not provide significant improvement, we chose to illustrate the control-target trade-off for many random magnitudes $(\alpha_i)$. As can be seen in Fig. 4, per TV magnitudes may provide only a slightly better tradeoff. We conclude that learned magnitudes cannot sufficiently address the problem of degradation when applying many narrow TVs. Implementation details can be found in the App.D.

**Task Arithmetic in the Tangent Space Ortiz-Jimenez et al. (2024).**    A recent work suggests that TV arithmetic works partly because of weight disentanglement. Namely, they claim that different TV mainly change different parameters in the model. The authors propose a method to encourage weight disentanglement through learning TVs in the Neural Tangent Space Ortiz-Jimenez et al. (2024); Jacot et al. (2018). We investigate using this method as another option to mitigate the degradation of the control task performance. In Fig. 7 we plot the control task performance and the erasure task performance for different numbers of combined TVs. We find that the Tangent Space TV method does not mitigate the linear degradation in the control performece.

**Joint TV Training.**    A possible simple modification to the TV setting is to train a single TV aimed at jointly performing multiple target tasks together, instead of training $N$ vectors individually and combining them later. While this technique may convey some desired properties of the TV technique, like the the option to control and reverse the edit amplitude; it does not allow other benefits like combining TV from different sources. To evaluate this technique potential to better preserve a control tasks performance, we train a TV on many tasks together, and compare the control-target performance trade-off it provides to that of the TV baseline techniques. We can see in Fig. 6 (Co-Training) that unlike the previous solutions explored in this section, this technique does provide a somewhat better trade-off, and we recommend it as a practical solution for somewhat mitigating control task degradation when possible. Yet, this solution is still not enough to allow the application of TV-based erasure for large values of $N$.

## 5   ADAPTIVE TASK VECTOR SELECTION

In the previous section, we found that existing solutions cannot sufficiently preserve control task performance under multiple task edits. In this section we present a possible solution. Since combining a large number of TVs significantly degrades the control task performance, we aim to decide during inference time which TV should be applied for a given sample. Our main idea is that for a given prompt, different TVs will differently affect the denoising process; and that this difference can be tracked during inference time.

**Adaptive test-time selection of Task Vector.**    Our technique relies on a simple assumption: a Task Vector that is irrelevant to a given generation would tend to produce a smaller semantic difference in the output image compared to relevant one TV. Identifying irrelevant Task Vectors would enable us not to apply them to a given generation, and would prevent the degradation they cause the control task accuracy. Therefore, we first apply each Task Vector edit on its own to find which TVs are relevant.

Namely, we first generate an image using the original model $G_{\theta_0}$ and the text prompt $p$:

$$X_0 = G_{\theta_0}(p) \qquad (10)$$

---

**Algorithm 1** Adaptive Task Vector Edits for Diffusion Models

---

**Input:** Prompt $p$, set of Task Vectors $\{\tau_i\}_{i=1}^{N}$, original diffusion model $G_{\theta_0}$ (with parameters $\theta_0$), similarity threshold $s_T$, denoising switching time step $t_{switch}$, TV magnitude $\alpha$
**Output:** Generated image with selective TV application
1: Generate baseline image: $X_0 = G_{\theta_0}(p)$
2: **for** each Task Vector $\tau_i$ **do**
3:     Initialize generation: $X_t = G_{\theta_0}(p, t)$    ▷ Generate up to time step $t_{switch}$ w. original model
4:     Continue generation from $t$ with the edited model: $X_i = G_{\theta_0 + \alpha\tau_i}(X_t)$
5:     Compute similarity: $s_i = sim(X_0, X_i)$                    ▷ Using CLIP embeddings
6: **end for**
7: Initialize combined TV: $\tau_{comb} = 0$
8: **for** each similarity score $s_i$ **do**
9:     **if** $s_i > s_T$ **then**
10:         $\tau_{comb} = \tau_{comb} + \alpha\tau_i$                    ▷ Add relevant TVs
11:     **end if**
12: **end for**
13: $X_{final} = G_{\theta_0 + \tau_{comb}}(X_t)$
14: **return** $X_{final}$

---

Figure 5: ***Mid-process Selective TV allows to select only the TV edits that are relevant to the prompt at hand. (Top row)*** *The full diffusion process with the original unedited model. (**Middle row**) The diffusion process, when editing the model at time $t = 30$ with a relevant TV edit (subtracting "Van Gogh"). The final generation has low similarity to the image generated by the original model. (**Bottom row**) The diffusion process, when editing the model at time $t = 30$ with a less relevant TV edit (subtracting "Killian Eng"). Accordingly, the final generation has a higher similarity to the image generated by the original model.*

Table 1: ROC AUC values for detecting relevant TV for image generation with different prompts. In each prompt, the different examined artistic styles are inserted into the place denoted by #. Additional results can be found in the Appendix.

| Prompt | # | A # painting of a cat | A biblical scene by # | #-themed still life |
|---|---|---|---|---|
| ROC AUC | 0.946 | 0.861 | 0.911 | 0.908 |

Next, we generate $N$ images, using the $N$ given TVs. We start with the original model, and switch to the edited model at time $t$ of the de-noising process:

$$\{X_i\} = G_{\theta_i(t)}(p), \quad \theta_i(t) = \begin{cases} \theta_0, & \text{if } t < t_{\text{switch}} \\ \theta_0 + \alpha \cdot \tau_i, & \text{if } t \geq t_{\text{switch}} \end{cases} \tag{11}$$

Finally, we examine the semantic similarity of each of the generated images $\{X_i\}$ with the baseline image $X_0$. We evaluate the similarity using cosine similarity of CLIP embeddings, noted by $sim$:

$$s_i = sim(X_0, X_i) \tag{12}$$

As relevant TVs are expected to change the output more significantly, we expect the similarity score $s_i$ to be smaller for the TVs that edit concepts relevant to the generation.

**Time-selective Task Vectors edits.** While we expect irrelevant TVs will only have a small effect on the generated image, applying such TVs before the first diffusion denoising step may still change the output image significantly. This happens because the generation does not depend only on the model but also on the initial noise. Any small intervention at an early timestep changes the initial patterns formed from that noise, and therefore is somewhat similar to re-seeding the noise pattern. One may apply the TV edits only at the end of the denoising process, but then it may not have a significant enough impact on the output image since all of the high-level image features are already formed. Therefore, we apply each TV edit in the middle of the denoising process at some time $t_{switch}$. See Fig. 5 for illustration and the Tab.2 for empirical ablation.

**Evaluation.** We begin by demonstrating that our method can identify the relevant TV among a selection of prompts. We inspect 6 artistic styles—*(1) Ajin: Demi-Human, (2) Kelly McKernan, (3) Kilian Eng, (4) Thomas Kinkade, (5) Tyler Edlin,* and *(6) Van Gogh* — and train a TV for each of them. We generate an image with prompts related to each artistic style and evaluate our method's ability to identify the relevant TV associated with this style (Tab. 1). As users may tune the control-erasure trade-off by changing the threshold for the inclusion of a given TV, we evaluate our TV selection method independently from this threshold by using the ROC-AUC metric. Our evaluation shows a significant ability to identify relevant TV with some prompts and is only somewhat indicative when using other prompts. Yet, even an imperfect ROC AUC score allows us a to discriminate between completely irrelevant TVs and TVs that might be relevant, significantly reducing the number of irrelevant TVs we would need for a given generation.

To illustrate the potential of our technique for achieving a better control-target trade-off we plot in Fig.6 the trade-off between the generation accuracy on the target concept (the target task is erasure, so lower is better) and the control accuracy of generating unrelated concepts which we aim to erase. Both accuracies are measured using the CLIP similarity between the text prompt and the generated image. We compare our method in Fig.6 to two baselines: (i) simple TV addition (ii) Co-training a joint TV Training (as described in Sec.4). We can see that while co-training mitigates only a bit the adversarial effect of TV subtraction, our method preserves the control accuracy much better. The implementation details for this experiment can be found in App.D.

**Ablation study:**

*Edit Time in the denoising process:* As mentioned earlier, during TV selection, we suggest starting the generation with the original model $G_{\theta_0}$ and moving to the edited model $G_{\theta_0 + \alpha \cdot \tau_i}$ in the middle of the diffusion denoising process at time $t_{switch}$. We ablate different choices of $t_{switch}$ in Tab. 2 and find that an intermediate timestep intervention is indeed beneficial.

*TV Edit Strength:* A second factor that might affect our ability to identify the relevant TVs is the magnitude $\alpha$ with which we inspect the different TVs. We ablate this choice in Tab. 3.

Table 2: ROC AUC values for detecting relevant TV for image generation with intervention timestamps (out of 50 denoising steps)

| $t_{switch}$ | 0 | 10 | 20 | 30 | 40 | 50 |
|---|---|---|---|---|---|---|
| ROC AUC | 0.889 | 0.922 | 0.946 | 0.874 | 0.829 | 0.500 |

Table 3: ROC AUC values for detecting relevant TV for image generation with TV scaling values.

| $\alpha$ | 0.5 | 1.0 | 2.0 | 3.0 | 4.0 | 5.0 |
|---|---|---|---|---|---|---|
| ROC AUC | 0.732 | 0.904 | 0.946 | 0.889 | 0.661 | 0.548 |

# 6 DISCUSSION AND LIMITATIONS

**Relation to input and output filtering methods.** Robust edits to generative models is likely to combine many components, with TV edits being just one of them just one of them. One of the advantages of TV edits is their inference time controllability Gandikota et al. (2023). Additionally, as Task Vectors are defined based on a target task; and therefore may be more robust than classifiers Pham et al. (2024). Yet, additional techniques to selectively applit edits may also help in reducing the degradation effect Task Vectors have on the control accuracy.

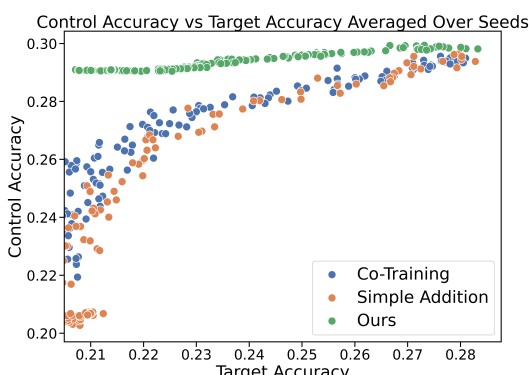

Figure 6: *Our method of inference-time selection of TV allows us to reduce the accuracy of the concept we wish to erase (Target Accuracy, lower is better) while maintaining the generation quality of other concepts (Control Accuracy, higher is better).* We plot the Target Accuracy and Control Accuracy trade-off for our method, simple TV arithmetic, and joint training of different tasks.

**Runtime Considerations for Adaptive Task Vector Subtraction.** The method presented in Sec.5 might exhibit a somewhat slower runtime with respect to the usual diffusion process. Diffusion models tend to be large and may take a long time to load and unload from GPUs. That is, even though we do not run the entire diffusion process for every generator, the GPU memory considerations may induce a runtime bottleneck for the presented algorithm. Therefore, deploying this algorithm at scale may be more efficient: when many queries are being executed in parallel. In this case,the TVs can be tested in large batches, spreading out the GPU bottleneck across many machines.

**Extension to Other Models and Edit technique.** Extension of our study to other generative models, such as LLMs is an exciting future direction . In addition, other inference time edit methods in text-to-image model are likely to have multi-task interactions as well, are may suffer from similar issues applying many edits at the same time.

# 7 CONCLUSION

We started this study by exploring the effects of multiple task vector interactions on a model's control task performance. Motivated by this, we turned to investigating how model degradation may be mitigated when subtracting many different task vectors from the same model. We explored a large variety of methods and found that simple or existing technique do not sufficiently mitigate the degradation of the model on tasks unrelated to the applied TV edits. Therefore, we sugguested an adaptive technique that finds the relevant TV to be applied to a diffusion model at the inference time. Finally, we evaluate our suggested method and find it is effective in mitigating the degradation generations unrelated to the applied TV edits.

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

REPRODUCIBILITY STATEMENT

We include implementation details for our analysis experiments and code for the suggested method.

IMPACT STATEMENT

Studying edits to foundation models can impact society in various ways. On one hand, it might enhance the controllability of these models and reduce their potential to cause social harm. On the other hand, improving their quality could introduce new harmful capabilities. This work, however, focuses on the fundamental interactions between tasks rather than any specific capabilities. Therefore, we do not believe its impact significantly differs from that of the majority of studies investigating foundation models.

## A   MULTI-TASK INTERACTION WITH TASK ARITHMETIC IN THE TANGENT SPACE

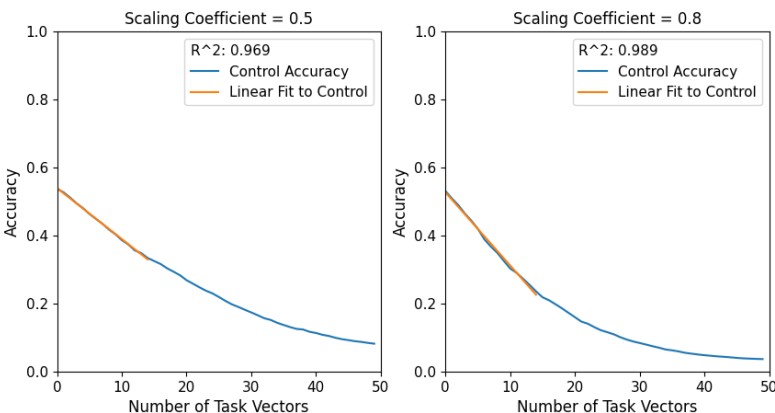

Figure 7: **The control task degrades linearly with increasing the amount of subtracted TVs also for Task Arithmetic in the Tangent Space TVs.** *Control task accuracy (50-classes CIFAR classification) of a CLIP backbone as function of the number of TV edits applied with a fixed magnitude each, even when using the tangent space technique technique by Ortiz-Jimenez et al. (2024). When editing an increasing number of TVs, the control task performance linearly degrades up to a significant amount of vectors. We examine two different scaline coefficients depicted in the title of each figure.*

# B  PAIR-WISE INTERACTION TYPE FOR A CLIP CLASSIFIER

Non-linearity score: 0.007
Total angle: 20.3
No. of layers w. $\phi_i > 75$deg: 6

Non-linearity score: 0.052
Total angle: 83.4
No. of layers w. $\phi_i > 75$deg: 50

Non-linearity score: 0.104
Total angle: 89.2
No. of layers w. $\phi_i > 75$deg: 59

Non-linearity score: 0.093
Total angle: 90.0
No. of layers w. $\phi_i > 75$deg: 158

Figure 8: **The pair-wise interaction type of different Task Vectors correlates with the semantic similarity of the tasks.** For each of Task Vectors we report (i) The permanence heatmap based on the two TV edit magnitudes. (ii) The non-linearity score defined as the average normalized difference between diagonal and off-diagonal (edge) elements in the similarity matrix (iii) The total angle between the Task Vector, and (iv) The number layer with internal angle of above 75 degrees threshold between the two Task Vectors. As we can see, the total angle and number of layers above the threshold correlate with non-linearity as seen in the graph and quantified by our non-linearity score.

# C  PAIR-WISE INTERACTION TYPE FOR A STABLE DIFFUSION

We include heatmaps for the interaction of TV applied to a stable diffusion model in Fig.9. We note that class-specific TV addition degrades the model similarly to TV subtraction.

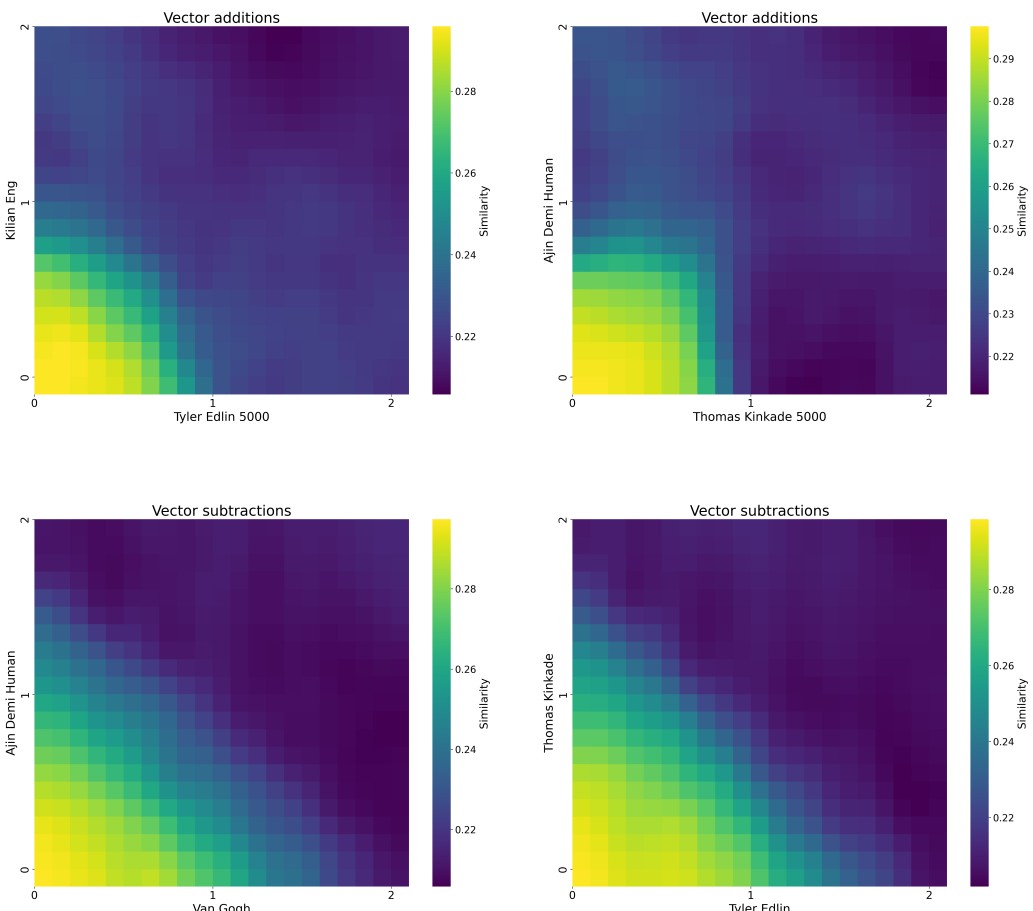

Figure 9: **Interaction maps for positive (addition) and negative (subtraction) TV edits to a stable diffusion model.**

## D  IMPLEMENTATION DETAILS

**Scaling Under Multiple Task Interactions.** To generate the 50 TVs necessary for this experiment, we finetuned the classification head of a classifier with a CLIP ViT-B-32 backbone independently on 50 different CIFAR-100 classes, each for 3 epochs with a batch size of 128 and learning rate of 1e-5.

**Non-Linear TV Combination & Learnable Task Vector weights.** In each of these experiences, we examine 5 Task Vectors, fine-tuned for a single class out of 5 CIFAR100 classes (out of the first 50). We use accuracy on the last 50 classes as the control task. We scan the magnitude hyperparameter sporadically to generate relevant values for control task performance.

**Control Task for Stable Diffusion.** We use the generation quality (measured by CLIP) as the control task accuracy for the SD1.4 foundation model: "Alphonse Mucha", "H.R. Giger", "Gustav Klimt", "Hayao Miyazaki", "M.C. Escher", "Yoshitaka Amano", "Salvador Dalí", "James Gurney", "Jean Giraud (Moebius)", "John Singer Sargent", "airplane", "automobile", "bird", "cat", "deer", "dog", "frog", "horse", "ship", "truck".

**Task Vectors.** We train all task vectors using 5000 epochs of a standard SD1.4 finetuning procedure, using 10 images for each reported concept.

**Adaptive Task Vector Selection.** For Fig.6 we examine these tasks pairs: ("ajin demi human", "kelly mckernan"), ("kilian eng", "thomas kinkade"), ("thomas kinkade", "tyler edlin"), ("tyler edlin", "van gogh"), ("van gogh", "ajin demi human"), ("kelly mckernan", "kilian eng").

We use a single concept TV for "Our" and "Simple addition", and train task vectors for concept pair in "Co-Training". The presented results are averaged across the different pairs that we may use.

