# OpenReview forum: "Multi-Concept Editing Using Task Arithmetic"
_ICLR.cc/2025/Conference — ICLR 2025 Conference Withdrawn Submission_

### Official Review · Reviewer_QDni · 2024-10-20

**Soundness:** 1
**Presentation:** 1
**Contribution:** 2
**Rating:** 5
**Confidence:** 3

**Summary:**

This paper studies the trade-off between target task performance and control task performance when task vectors are applied to a diffusion-based image editing model. It shows that existing solutions can not preserve control task performance well under multiple task edits. Addressing this issue, this paper introduces a task vector selection strategy to adaptively apply only the relevant task vectors to a given prompt in the diffusion-based image editing model.

**Strengths:**

1) The proposed TV selection algorithm is simple and clear.

2)  Experiment results validate that the proposed method can reduce the accuracy of the undesired concept while maintaining the control task performance.

**Weaknesses:**

1) The introduction is confusing.
The first few paragraphs introduce an observation that accuracy degradation accumulates with the total magnitude of the task vectors. However, the paragraph starting from L75 suddenly brings up the diffusion process, which is irrelevant to previous discussions.

2) The motivation for applying task vectors to diffusion models is unclear.
Using LoRA as in Gandikota et al., 2023 appears to be more efficient. What are the problems in existing editing techniques and what are the benefits of using task vectors?

3) The experiment results are not convincing.
Lack of comparison with existing methods such as Gandikota et al., 2023.

**Questions:**

1) In the introduction section, how is the diffusion process in L75 related to previous discussions? Isn't the observation of task vectors trade-off applicable to most models? Or is it specific to the diffusion model?

2) Regarding the motivation for using task vectors, what are the problems in existing editing techniques and what are the benefits of task vectors, especially when compared to LoRA?

3) There is only one visualized example of the edited results. I suggest the author provide more visualizations.

4) I suggest the authors show quantitative and qualitative comparisons with existing image editing methods.

---

### Official Review · Reviewer_2Ai1 · 2024-11-03

**Soundness:** 2
**Presentation:** 2
**Contribution:** 2
**Rating:** 5
**Confidence:** 3

**Summary:**

This work studies the impacts of interactions among task vectors when multiple task vectors are combined. The authors first discuss the linear degradation of model performance when multiple task vectors are applied to a single image classification model. This phenomenon is explained through a simple mathematical model using a normal distribution. Then, different methods to overcome this phenomenon are discussed. In the end, an adaptive task vector selection algorithm is proposed for the diffusion model to control performance when multiple task vectors are applied.

**Strengths:**

* Applying multiple task vectors to a single model seems interesting.
* The linear degradation of model performance when multiple vectors are combined is intriguing.

**Weaknesses:**

* The paper lacks details regarding the datasets used. For image classification, are only SUN397, MNIST, and SVHN utilized? If so, the results are not convincing enough.
* The simple mathematical model explains the linearity of task vector strength when multiple vectors are combined. However, the relationship between task vector strength and accuracy may not be linear unless $\alpha_c$ is small.
* Missing experimental details such as value of $\alpha_c$ in L257.
* The paper dedicates a large portion to demonstrating the combination of multiple task vectors on an image classification model, but it later suddenly switches to a diffusion model, which seems odd. What is the corresponding adaptive test-time selection of task vectors for the image classification model?

Minor issue: Citations within the text are strange, check the formatting instruction.

**Questions:**

* In L257, what is the value of $\alpha_c$?
* How does this approach compared with model reprogramming?

---

### Official Review · Reviewer_1wEc · 2024-11-04

**Soundness:** 3
**Presentation:** 2
**Contribution:** 3
**Rating:** 3
**Confidence:** 3

**Summary:**

This paper studies the interactions of task vectors in a multi-edit setting for image classifiers and diffusion models, finding that quality degrades as more edits are applied.
The authors also propose a new method that adaptively applies only the relevant Task Vectors, enabling effective integration of multiple TV edits while significantly mitigating quality degradation.

**Strengths:**

1. The paper presents a novel study on how multi-task-vector interactions affect the control task performance of a model.
2. The paper provides a comprehensive survey of existing editing methods based on TV techniques.

**Weaknesses:**

1. The paper lacks visual examples of multi-editing results, making it harder to assess the exact editing performance.
2. The evaluation is insufficient, as it includes only six artistic styles, limiting its ability to reflect editing performance across various tasks.

**Questions:**

See the Weaknesses Section.

---

### Official Review · Reviewer_E79Z · 2024-11-04

**Soundness:** 3
**Presentation:** 3
**Contribution:** 2
**Rating:** 5
**Confidence:** 4

**Summary:**

The manuscript targets the problem of model editing on multiple tasks, over the concept of task vectors where they serve as the model weights combined with the original model o achieve the targeted task. Preliminarily, the authors first provide an analysis on how does the current approaches on applying multiple-concept editing perform, by first introducing different kinds of interactions between task vectors (linear and non-linear). In the first part of the provided analysis, authors attempt to quantitively assess the influence between them over changing performance on the desired tasks and the strength of the applied task vector. Following, the authors consider different strategies on combining the subjected task vectors. Throughout the presented experiments, the authors show how the current approaches fell short over classification oriented multi-concept editing, where they also provide discussions and background on each of the existing approaches for multi-task arithmetic.

As the second part of the manuscript, which also serves as the second contribution, they propose an inference time solution for performing multi-task arithmetic on diffusion models. In the proposed algorithm, the authors introduce an heuristic based multi-task arithmetic operation, that relies on the similarity between the images generated by edited and non-edited models. Following, they assess the performance of detecting the relevant task vectors over 6 different artistic styles.

**Strengths:**

- The authors provide a valuable analysis on the task of multi-task model editing, and effectively address the existing challenges.
- Authors provide a detailed overview on the existing approaches for the solution of the task and show their shortcomings quantitatively.
- Paper introduces an algorithm to perform multi-task arithmetic on diffusion models given a set of trained task vectors, by introducing a image pair similarity based heuristic.

**Weaknesses:**

- The paper introduces an algorithm that identifies required task vectors and performs multi-task arithmetic accordingly. However, the authors have no analysis on the overall effect of the applied task vectors on the image. They claim that the existing methods cannot effectively combine multiple task vectors. If the proposed algorithm is to solve this problem with an heuristic, they also need to provide an analysis on the task performance. In addition, the task performance should also be shown qualitatively with additional examples as they propose the algorithm on diffusion models.
- The first part of the paper completely investigates the multi-task arithmetic problem over the task of classification. However, the proposed algorithm targets diffusion models completely. These two parts does not seem completely connected, and the provided analysis on multi-task editing should be extended with experiments on diffusion models also.
- The experiments on diffusion models are limited with artistic styles, where each task vector will involve same set of features. However, it is still a question that needs to be answered that if the proposed method can handle different kinds of task vectors at the same time and is it limited to artistic styles only.
- The details of how the authors train the task vectors is not clear. Even if an external method is used, it has to be mentioned explicitly.
- Diffusion models encode information with different granularity over iterating timesteps[1]. As the authors also select a timestep to perform the arithmetic task, they should also discuss how the timestep selection effects the performance of task vectors (ideally with a visual analysis and demonstrations).
- Details of the baseline models that the experiments are conducted on are not provided (i.e. the diffusion model used).

[1] Prompt-to-Prompt Image Editing with Cross-Attention Control, Hertz et. al., 2022.

**Questions:**

- How does the proposed algorithm behave on task vectors on visual tasks other than stylization (e.g. semantic editing)? In the current form, the applications seem limited and cannot be generalized to many of the tasks.
- Is it possible to conduct an analysis on combining the task vectors in diffusion models and how current approaches perform? Otherwise, it is not clear what the baseline approach is, and how existing approaches perform on the task that the algorithm is proposed on.
- Other than identifying the task vectors, how does the proposed algorithm behave in terms of visual coherence? How does the image quality change? How well does it perform the task? This aspect is crucial for diffusion models as it helps readers to assess how visually pleasing the proposed algorithm is, in addition to how well the tasks are identified.
- What are the algorithms used to train the task vectors?

---

### Author Response · Authors · 2024-11-15

We thank all the reviewers for their dedicated efforts. Given the status of the reviews, we will revise the manuscript and submit it to another conference.

---

### Note · Authors · 2024-11-15

I have read and agree with the venue's withdrawal policy on behalf of myself and my co-authors.